# Predicting human decision making in psychological tasks with recurrent neural networks

**Baihan Lin** [1,2,3]*, **Djallel Bouneffouf** [4], **Guillermo Cecchi** [5]

1 Department of Systems Biology, Columbia University, New York, NY, United States of America,
2 Department of Neuroscience, Columbia University, New York, NY, United States of America, 3 Department of Psychology, Columbia University, New York, NY, United States of America, 4 Department of Artificial Intelligence Foundations, IBM Thomas J. Watson Research Center, Yorktown Heights, NY, United States of America, 5 Department of Healthcare and Life Sciences, IBM Thomas J. Watson Research Center, Yorktown Heights, NY, United States of America

* baihan.lin@columbia.edu

**Data Availability Statement:** The data and codes to reproduce all the empirical results can be accessed and reproduced at https://github.com/doerlbh/HumanLSTM.

## Abstract

Unlike traditional time series, the action sequences of human decision making usually involve many cognitive processes such as beliefs, desires, intentions, and theory of mind, i.e., what others are thinking. This makes predicting human decision-making challenging to be treated agnostically to the underlying psychological mechanisms. We propose here to use a recurrent neural network architecture based on long short-term memory networks (LSTM) to predict the time series of the actions taken by human subjects engaged in gaming activity, the first application of such methods in this research domain. In this study, we collate the human data from 8 published literature of the Iterated Prisoner's Dilemma comprising 168,386 individual decisions and post-process them into 8,257 behavioral trajectories of 9 actions each for both players. Similarly, we collate 617 trajectories of 95 actions from 10 different published studies of Iowa Gambling Task experiments with healthy human subjects. We train our prediction networks on the behavioral data and demonstrate a clear advantage over the state-of-the-art methods in predicting human decision-making trajectories in both the single-agent scenario of the Iowa Gambling Task and the multi-agent scenario of the Iterated Prisoner's Dilemma. Moreover, we observe that the weights of the LSTM networks modeling the top performers tend to have a wider distribution compared to poor performers, as well as a larger bias, which suggest possible interpretations for the distribution of strategies adopted by each group.

## 1 Introduction

Predictive modeling involves the use of statistics to predict outcomes of "unseen" data, i.e. not used in model parameterization, in real world phenomena, with wide applications in economics, finance, healthcare, and science. In statistics, the model that closely approximates the data generating process might not necessarily be the most successful method to predict real world

**Funding:** The author(s) received no specific funding for this work.

**Competing interests:** The authors have declared that no competing interests exist.

outcomes [1, 2]. Yarkoni & Westfall [3] argue that the near-total focus of the field of psychology on explaining the causes of behavior have little or unknown capability to predict future human behaviors with any appreciable accuracy, despite the intricate theories of psychological mechanism these research have endowed. Methods like regression and other mechanistic models, even with high complexity, can still be outperformed by biased and psychologically implausible models, due to overfitting. In this work, we aim to bridge this gap, by providing machine learning methods to accurately predict game-based human behavior.

While useful for the researchers to breakdown neuropsychologically interpretable variables, these analyses only provide very constrained predictive guidelines, and fall short to modeling more complicated real-world decision making scenarios such as social dilemmas. As a popular framework to expose tensions between cooperation and defection in a game-like manner, the Iterated Prisoner's Dilemma [4] has been studied by computer scientists, economists, and psychologists with different approaches. Beyond cognitive modeling of the effects of game settings and past experiences on the overall tendency to cooperate (or invest in the monetary games) [5–7] proposed a logistic regression model to directly predict individual actions during the Iterated Prisoner's Dilemma. This logistic regression model is also the state-of-the-art in predicting action sequences in this task.

As another commonly used game-based task, albeit non-social, the Iowa Gambling Task [8] is usually modeled as a synthesis of various psychological processes and cognitive elements [9, 10]. In the Iowa Gambling Task, the participant needs to choose one out of four card decks (named A, B, C, and D), and can win or lose money with each card when choosing a deck to draw from [8]. The challenge of this kind of game in computational modeling is that the reward payoffs of each action arms are not necessarily Gaussian: all decks have a consistent wins and variable losses where one of the popular schemes, the scheme 1 [11], has a more variable losses for deck C than another one, the scheme 2 [12].

In both settings, one active line of research is to clone and simulate behavioral trajectories with reinforcement learning models that incorporate learning-related parameters inspired by the neurobiological priors of the human brain [13–16]. While offering discriminative and interpretable features in characterizing the human decision making process, these reinforcement learning models exhibit limited capability to predict the human action sequences in complicated decision making scenarios such as the Iterated Prisoner's Dilemma [17].

We propose in this paper to study decision-making based on the well-known long short-term memory network (LSTM) to forecast the human behavior using experimental data from game-like psychological tasks including the Iowa gambling task and the Iterated Prisoner's Dilemma, to then endeavor to interpret the resulting models, under the premise that good prediction is a requisite to warrant interpretation. To the best of our knowledge, this is the first study that applies recurrent neural networks to directly predict sequences of human decision making process. We believe this work can facilitate the understanding of how human behave in online game setting.

A preliminary version of this work was presented at the International Joint Conference on Neural Networks [18].

## 2 Materials and methods

### 2.1 Long-short-term-memory networks

Recurrent neural networks are a class of artificial neural networks that captures a notion of time. It has both conventional edges that map from input nodes to the recurrent nodes, and recurrent edges that map recurrent nodes across adjacent time steps, including cycles of length

one that are self-connections from a node to itself [19]. It can be formulated as:

$$\mathbf{h}_t = \sigma(W_{hx}\mathbf{x}_t + W_{hh}\mathbf{h}_{t-1} + \mathbf{b}_h) \tag{1}$$

where at time $t$, the recurrent layer receives input $\mathbf{x}_t$ and computes the neurons' hidden states $\mathbf{h}_t$ given the input $\mathbf{x}_t$, last hidden states $\mathbf{h}_{t-1}$ and the parameters including a kernel $W_{hx}$, a recurrent kernel $W_{hh}$ and a bias term $\mathbf{b}_h$. The activation $\sigma$ can be in different forms, with sigmoid function to be a conventional choice. The dynamics of the recurrent network can be considered as a deep neural network by unfolding the computing graph into layers with shared weights. Then we can train the unfolded network across many time steps with backpropagation with algorithm slike Backpropagation through time (BPTT) [20].

Long Short Term Memory (LSTM) was later introduced to overcome the vanishing gradient problem of recurrent networks [21]. The model resembles traditional recurrent neural networks, but introduces a series of gating mechanisms to adaptively keep a memory. It introduces four additional variables, the input gate $\mathbf{g}$, the input state $\mathbf{i}$, the forget gate $\mathbf{f}$, and the output gate $\mathbf{o}$. The full formulations are as follows:

$$\mathbf{g}_t = \phi(W_{gx}\mathbf{x}_t + W_{gh}\mathbf{h}_{t-1} + \mathbf{b}_g) \tag{2}$$

$$\mathbf{i}_t = \sigma(W_{ix}\mathbf{x}_t + W_{ih}\mathbf{h}_{t-1} + \mathbf{b}_i) \tag{3}$$

$$\mathbf{f}_t = \sigma(W_{fx}\mathbf{x}_t + W_{fh}\mathbf{h}_{t-1} + \mathbf{b}_f) \tag{4}$$

$$\mathbf{o}_t = \sigma(W_{ox}\mathbf{x}_t + W_{oh}\mathbf{h}_{t-1} + \mathbf{b}_o) \tag{5}$$

$$\mathbf{s}_t = \mathbf{g}_t \odot \mathbf{i}_t + \mathbf{s}_{t-1} \odot \mathbf{f}_t \tag{6}$$

$$\mathbf{h}_t = \phi(\mathbf{s}_t) \odot \mathbf{o}_t \tag{7}$$

where the activation functions are either sigmoid $\sigma$ or tanh $\phi$, and $\odot$ is pointwise mulitplication. A cell state $\mathbf{s}_t$ is computed based on the previous cell state $\mathbf{s}_{t-1}$ the input gate $\mathbf{g}$, the input state $\mathbf{i}$ and the forget gate $\mathbf{f}$. Then, the hidden state of the LSTM layer is computed by pointwise multiplying the activated cell state with the output gate.

**2.1.1 The Iterated Prisoner's Dilemma (IPD).** The Iterated Prisoner's Dilemma (IPD) can be defined as a matrix game $G = [N, \{A_i\}_{i \in N}, \{R_i\}_{i \in N}]$, where $N$ is the set of agents, $A_i$ is the set of actions available to agent $i$ with $\mathcal{A}$ being the joint action space $A_1 \times \cdots \times A_n$, and $R_i$ is the reward function for agent $i$. A special case of this generic multi-agent Iterated Prisoner's Dilemma is the classical two-agent case (Table 1). In this game, each agent has two actions: cooperate (C) and defect (D), and can receive one of the four possible rewards: R (Reward),

**Table 1. Payoff codes of the Iterated Prisoner's Dilemma.**

|  | Cooperate | Defect |
|---|---|---|
| Cooperate | (R, R) | (S, T) |
| Defect | (T, S) | (P, P) |

The rows and columns specify the action space of the agents as well as their corresponding payoff codes. At each round, each agent has two actions: Cooperate and Defect. Given the actions taken by one agent (the first column) and its opponent (the first row), they are each assigned one of the four possible rewards: R (Reward), P (Penalty), S (Sucker), and T (Temptation). The reward tuple reads (reward to the agent itself, reward to the agent's opponent).

P (Penalty), S (Sucker), and T (Temptation). In the multi-agent setting, if all agents Cooperates (C), they all receive Reward (R); if all agents defects (D), they all receive Penalty (P); if some agents Cooperate (C) and some Defect (D), cooperators receive Sucker (S) and defector receive Temptation (T). The four payoffs satisfy the following inequalities: $T > R > P > S$ and $2R > T + S$. The Prisoner's Dilemma is a one round game, but is commonly studied in a manner where the prior outcomes matter to understand the evolution of cooperative behaviour from complex dynamics [22].

We collate human data comprising 168,386 individual decisions from many human subjects experiments [5, 6, 23–28] that use real financial incentives and transparently convey the rules of the game to the subjects. As a a standard procedure in experimental economics, subjects anonymously interact with each other and their decisions to cooperate or defect at each time period of each interaction are recorded. They receive payoffs proportional to the outcomes in the same or similar payoff as the one we used in Table 1.

**2.1.2 The Iowa Gambling Task (IGT).** The original Iowa Gambling Task (IGT) studies decision making where the participant needs to choose one out of four card decks (named A, B, C, and D), and can win or lose money with each card when choosing a deck to draw from [8], over around 100 actions. In each round, the participants receives feedback about the win (the money he/she wins), the loss (the money he/she loses), and the combined gain (win minus lose). In the Markov Decision Process setup, from initial state I, the player select one of the four deck to go to state A, B, C, or D, and reveals positive reward $r^+$ (the win), negative reward $r^-$ (the loss) and combined reward $r = r^+ + r^-$ simultaneously. Decks A and B by default is set to have an expected payout (-25) lower than the better decks, C and D (+25). For baselines, the combined reward $r$ is used to update the agents. There are two major payoff schemes in IGT. In the traditional payoff scheme, the net outcome of every 10 cards from the bad decks (i.e., decks A and B) is -250, and +250 in the case of the good decks (i.e., decks C and D). There are two decks with frequent losses (decks A and C), and two decks with infrequent losses (decks B and D). All decks have consistent wins (A and B to have +100, while C and D to have +50) and variable losses (summarized in Table 2, where scheme 1 [11] has a more variable losses for deck C than scheme 2 [12]).

The raw data and descriptions of Iowa Gambling Task can be downloaded at [29]. It consists of the behavioral trajectories of 617 healthy human subjects performing the Iowa Gambling Task. The data set consists of original experimental results from 10 different studies,

**Table 2. Payoff schemes of the Iowa Gambling Task.**

| Decks | win per card | loss per card | expected value | scheme |
|-------|-------------|---------------|----------------|--------|
| A (bad) | +100 | Frequent: -150 (p = 0.1), -200 (p = 0.1), -250 (p = 0.1), -300 (p = 0.1), -350 (p = 0.1) | -25 | 1 |
| B (bad) | +100 | Infrequent: -1250 (p = 0.1) | -25 | 1 |
| C (good) | +50 | Frequent: -25 (p = 0.1), -75 (p = 0.1), -50 (p = 0.3) | +25 | 1 |
| D (good) | +50 | Infrequent: -250 (p = 0.1) | +25 | 1 |
| A (bad) | +100 | Frequent: -150 (p = 0.1), -200 (p = 0.1), -250 (p = 0.1), -300 (p = 0.1), -350 (p = 0.1) | -25 | 2 |
| B (bad) | +100 | Infrequent: -1250 (p = 0.1) | -25 | 2 |
| C (good) | +50 | Infrequent: -50 (p = 0.5) | +25 | 2 |
| D (good) | +50 | Infrequent: -250 (p = 0.1) | +25 | 2 |

In this game, the subject needs to choose one out of four card decks (named A, B, C, and D), and can win or lose money with each card when choosing a deck for over around 100 actions. In each round, the subject receives feedback about the win (the money he/she wins), the loss (the money he/she loses), and the combined gain (win minus lose). Decks A and B by default is set to have an expected combined payout (-25) lower than the better decks, C and D (+25). All decks have consistent wins (A and B to have +100, while C and D to have +50) and variable losses, with different variabilities in the two schemes.

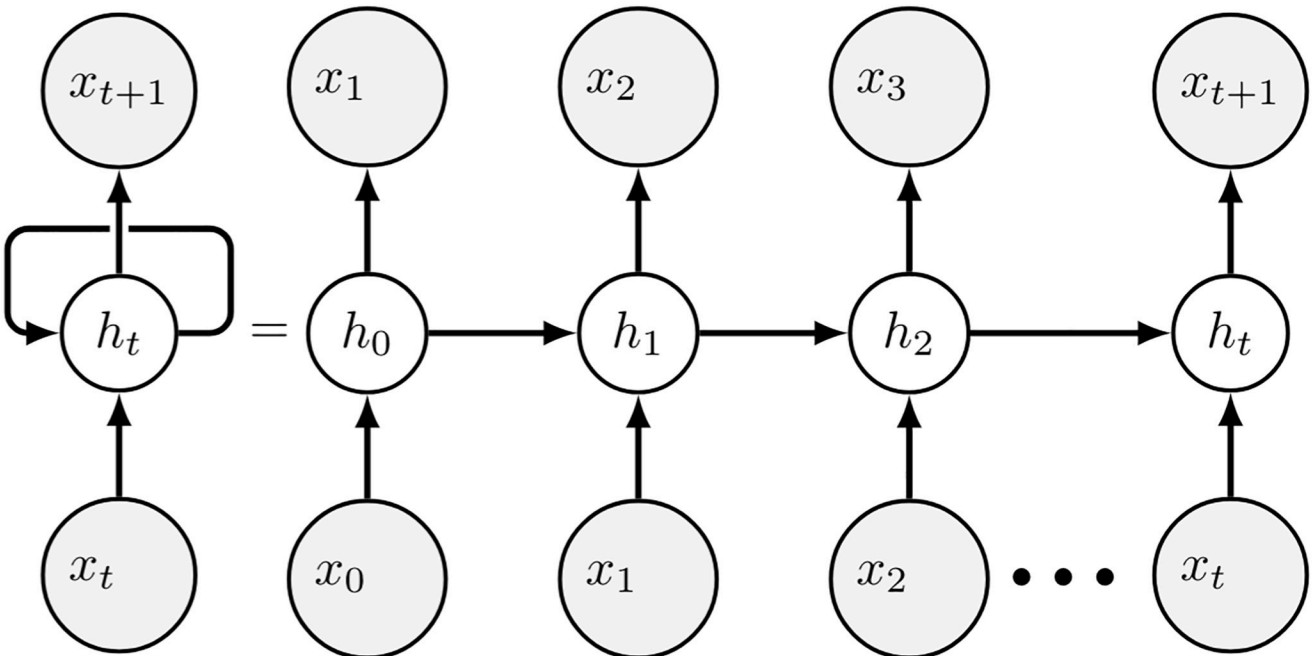

**Fig 1. Prediction framework with a recurrent network: At each time step $t$, the historical action from the last time step $x_t$ is fed into the recurrent neural networks as input, and the network outputs a predicted action $x_{t+1}$ for the following time step and updates its hidden state $h_t$.**

administrated with different lengths of trials (95, 100 and 150 actions). We pool all subjects together and truncate all 617 trajectories to have 95 actions each.

## 2.2 Prediction with recurrent networks

The prediction framework of the recurrent neural networks is illustrated in Fig 1: at each time step, the historical action from the last time step is fed into the recurrent neural networks as input, and the network outputs a predicted action for the following time step. In the single-agent setting (i.e. the human subject makes his or her decision without interactions with other players), the input and output of the recurrent neural networks both consist of the player's actions. In the multi-agent setting (i.e. the human subject makes his or her decision based on not only his or her own prior actions and rewards, but also the actions performed by other participants in the game), the input features consist of the actions performed by all parcipants in the game in the last time step. In both scenarios, we code the actions into a multi-dimensional one-hot representation before serving to the prediction network.

## 2.3 Network architecture and training procedures

We construct a neural network model that consist of multiple layers of LSTM networks with bias terms, followed by a ReLU activation function and a fully connected layer to map from the LSTM network output to a Softmax activation function, from which a prediction label is collected with an argmax operation. We implement the model in the standard library of PyTorch framework. We train the models for 200 and 400 epochs, respectively, for the Iterated Prisoner's Dilemma and the Iowa Gambling Task. We use the Adam [30] as the optimizer for the model and set the learning rate to be 1e-3 and a L2 regularization weight to be 1e-5. The

data and codes to reproduce the empirical results can be accessed and reproduced at https://github.com/doerlbh/HumanLSTM.

## 2.4 Evaluation of the Iterated Prisoner's Dilemma

Following the similar pre-processing steps as [7], we are able to construct a comprehensive collection of game structures and individual decisions from the description of the experiments in the published papers and the publicly available data sets. This dataset consists of behavioral trajectories of different time horizons, ranging from 2 to 30 rounds, but most of these experimental data only host full historical information of at most past 9 actions. We further select only those trajectories with these full historical information, which comprised 8,257 behavioral trajectories of 9 actions each for both players.

We compare with two baselines. The first baseline is a logistic regression taking into account a group of handcrafted features such as the game settings and historical actions [7]. It is reported as the state-of-the-art in the task of predicting human decision making in the task of Iterated Prisoner's Dilemma. Similar to the Iowa Gambling Task prediction, we include the standard vector autoregression model [31] as our second baseline. The order of the autoregression model is selected based on Akaike information criterion (AIC) [32]. Similar to the empirical evaluation in Iowa Gambling Task, here we still chose our prediction network to be a two-layer LSTM network with 5 neurons at each layers for the Iterated Prisoner's Dilemma prediction. We randomly split the dataset by 80/20 as the training set and the test set, and evaluated it with randomized cross-validation.

## 2.5 Evaluation of the Iowa Gambling Task

We compare our LSTM model with the standard vector autoregression model [31]. The order of the autoregression model is selected based on Akaike information criterion (AIC) [32]. The autocorrelation is trained on all available sequences in the training set. Same as our LSTM network, autoregression model takes a multi-dimensional one-hot-based feature tensor (of the previous time steps) as its observation window, and then outputs the next predicted action. For empirical evaluation, we chose our prediction network to be a two-layer LSTM network with 5 neurons at each layers. This is to showcase the effectiveness of our recurrent neural networks even if the parameter set is very small. We randomly split the dataset by 80/20 as the training set and the test set, and evaluated it with randomized crossvalidation.

# 3 Results

## 3.1 Predictive modeling of the Iterated Prisoner's Dilemma

We record the cooperation rate to evaluate how close the behavioral trajectory predicted by a model captures the ground truth of the human decision making sequence. As shown in Fig 2A, the autoregression model overestimates the cooperation rate of human decision making by a significant amount. The state-of-the-art model in this task, the logistic regression model [7] performs a better job than the autoregression model, but still falls short at capturing the subtle dynamics of human cooperation over time. Unlike the baselines, our LSTM model perfectly predicts the cooperation rate, although it is only trained to predict the individual actions instead of the cooperation rate.

We report the mean squared error (MSE) between the predicted sequences and the ground truth in order to understand the performance of the models in the individual sense. As shown in Fig 2B, LSTM has a lower average MSE of 0.12 across all prediction time steps, beating the two baselines, autoregression (0.18) and logistic regression (0.75), by a significant amount.

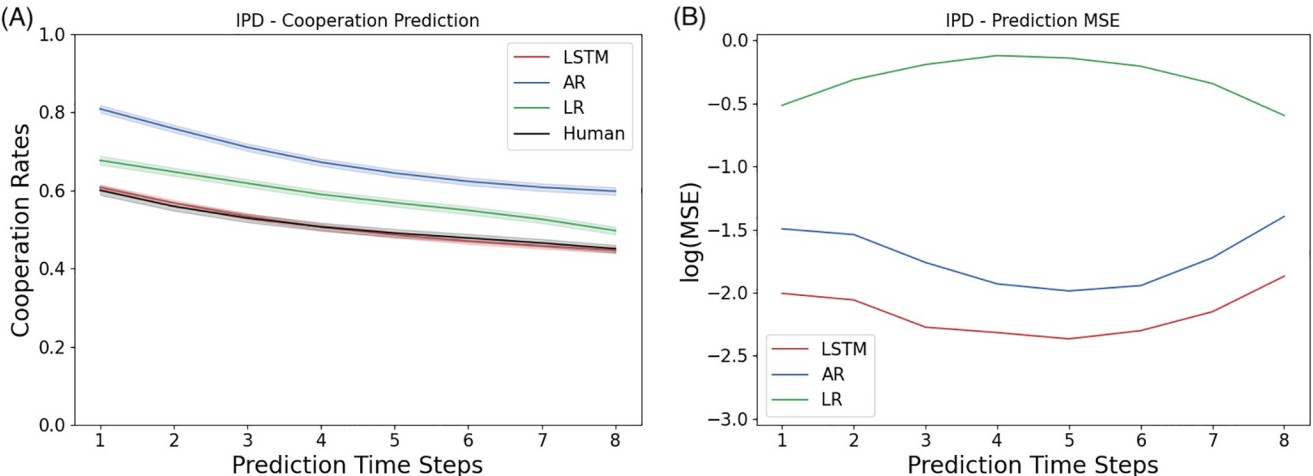

**Fig 2. Results for predicting the Iterated Prisoner's Dilemma: Shown here are the statistics computed from the prediction by a two-layer LSTM networks of five neurons at each layer.** The first time step is given as the prior, and we record the prediction of the next 8 time steps in a 9-step Iterated Prisoner's Dilemma game. (A) Cooperation Rate: The LSTM model characterizes human cooperation much better than the baselines in the Iterated Prisoner's Dilemma. (B) Individual Trajectories: The mean squared error of predicting indivdiual trajectories in the Iterated Prisoner's Dilemma.

### 3.2 Predictive modeling of the Iowa Gambling Task

We record this metric to evaluate how well the behavioral trajectory predicted by a model captures the ground truth of the human decision making sequence. As shown in Fig 3A, both the LSTM and the autoregression model capture the learning dynamics of the human subjects well.

As shown in Fig 3C, the overall MSE drops as the observation window increases (i.e. the model has seen more historical time series). The LSTM network predicts the individual trajectories better, with the lowest average MSE of 0.011 across all prediction time steps, beating the MSE by autoregression, 0.015, by a significant amount since the first few prediction time steps. As shown in Fig 3B, the LSTM network learns to mimic the overall learning trend of the Iowa Gambling Task in each decks.

### 3.3 Model complexity analysis

To investigate the effect of model complexity for the prediction tasks, we vary the number of neurons from 5, 10, 50 to 100, and vary the number of the LSTM layers from 1, 2 to 3. We replicate the aforementioned experiments on the Iterated Prisoner's Dilemma and the Iowa Gambling Task.

In the Iterated Prisoner's Dilemma, we observe that, the LSTM networks ranging from 1 layer of 5 neurons to 3 layers of 100 layers all predict the human cooperation rate very well, as shown in Fig 4.

Unlike the Iterated Prisoner's Dilemma, as shown in Fig 5, the LSTM networks ranging from 1 layer of 5 neurons to 3 layers of 100 layers vary in their similarity of the learning curve in choosing the better actions to that of the human data. In a close look at the actual prediction of each action dimension, we observe that across many models, there is a overall under-prediction of deck A and over-prediction of deck B, the two bad decks, as in the Fig 6.

### 3.4 Comparison of good and bad performers

To further investigate the interpretability of the trained LSTM networks, we subset the human subject data into two groups. The first group, which we call "top performers", are the players

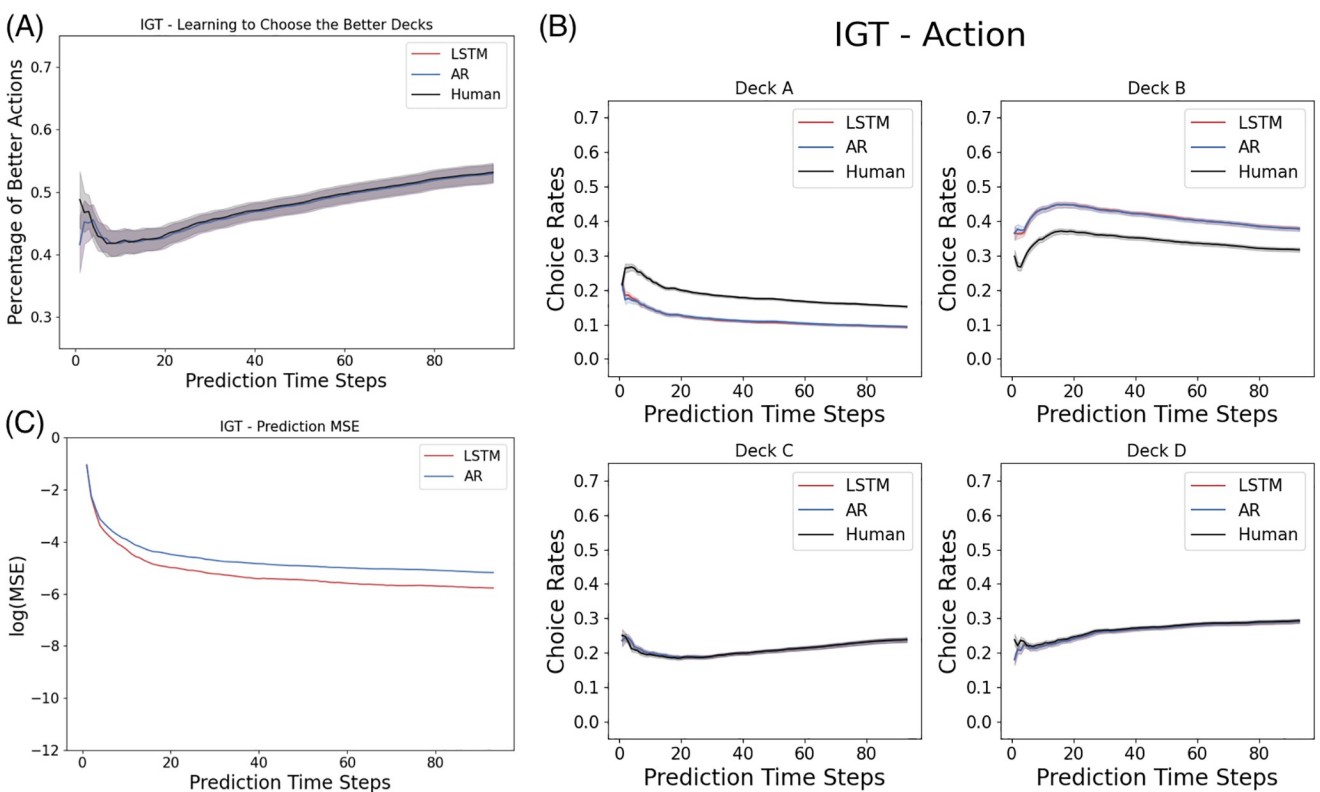

**Fig 3. Results for predicting the Iowa Gambling Task: Shown here are the statistics computed from the prediction by a two-layer LSTM networks of five neurons at each layer.** The first time step is given as the prior, and we record the prediction of the next 94 time steps in a 95-step Iowa Gambling Task game. (A) Learning Curve: Both the LSTM and autoregression model capture the learning dynamics of the human subjects in the Iowa Gambling Task as measure by the evolution of the rate of selection of the better actions. (B) IGT Prediction of LSTM and autoregression. (C) Individual Trajectories: The mean squared error of predicting individual actions in the Iowa Gambling Task.

that yield the top 25 percent scores in the games. The second, group, which we call "bottom performers", are the players that yield the bottom 25 percent scores in the games. As in the previous investigations, we choose our prediction network to be a two-layer LSTM network with 5 neurons at each layers. We train the networks for 100 epochs and 50 epochs, respectively, for the Iowa Gambling Task and the Iterated Prisoner's Dilemma. For each experiment, we trained a population of 100 randomly initialized instances of the LSTM networks and record their network weights.

We perform the (one-sample or two-sample) Kolmogorov-Smirnov test for goodness of fit [33] on the parameter weight distribution of the top and bottom performers.

In predicting the Iterated Prisoner's Dilemma, we present the distribution of the weights in the trained LSTM networks. We don't observe such a difference, as shown in Fig 7. These observation is supported by the Kolmogorov-Smirnov test: $W_{ih}^1$ (0.022, p = 0.041), $W_{hh}^1$ (0.009, p = 0.070), $b_{ih}^1$ (0.023, p = 0.217), $b_{ih}^1$ (0.008, p = 1.000) for layer 1; and $W_{ih}^1$ (0.011, p = 0.013), $W_{hh}^1$ (0.006, p = 0.424), $b_{ih}^1$ (0.024, p = 0.188), $b_{hh}^1$ (0.024, p = 0.188) for the layer 2.

In predicting the Iowa Gambling Task on the other hand, we observe that the weights of the top performers tends to have a wider distribution, and a bigger bias in the LSTM networks (Fig 8). These observation is supported by the Kolmogorov-Smirnov test: $W_{ih}^1$ (0.092, p = 1.77e-59), $W_{hh}^1$ (0.069, p = 4.46e-82), $b_{ih}^1$ (0.134, p = 5.04e-32), $b_{ih}^1$ (0.131, p = 1.63e-30) for layer 1; and $W_{ih}^1$ (0.078, p = 4.82e-107), $W_{hh}^1$ (0.059, p = 3.95e-60), $b_{ih}^1$ (0.082, p = 4.24e-12), $b_{hh}^1$ (0.074, p = 4.69e-10) for the layer 2.

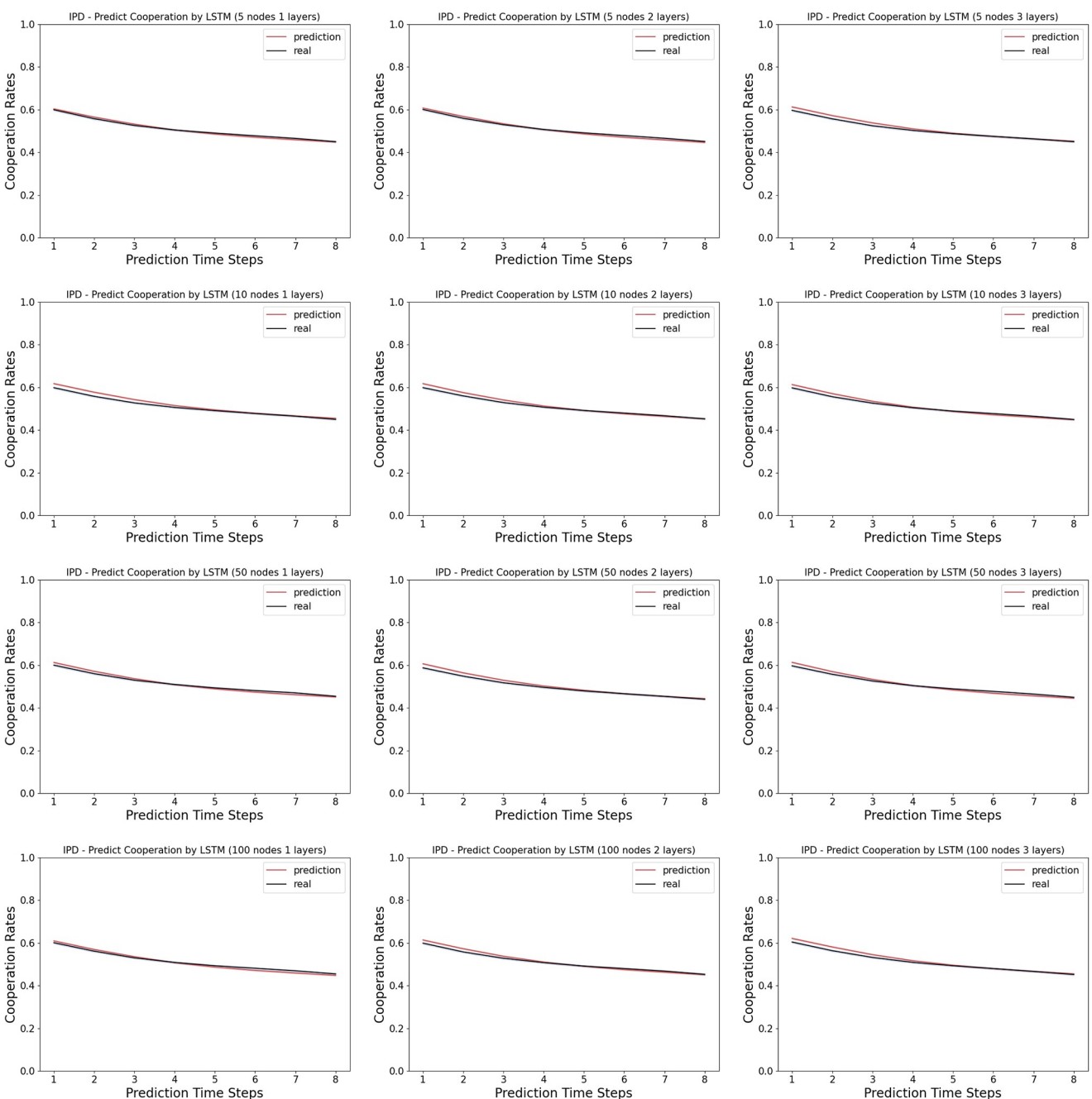

**Fig 4. Model complexity analysis in the Iterated Prisoner's Dilemma: Shown here are the cooperation rates computed from the prediction by the LSTM networks versus the real human data.** The first time step is given as the prior, and we record the prediction of the next 8 time steps in a 9-step Iterated Prisoner's Dilemma game. The columns indicates the number of layers in the LSTM networks, ranging from 1, 2 and 3. The rows indicates the number of neurons in each layer of the LSTM networks, ranging from 5, 10, 50 and 100.

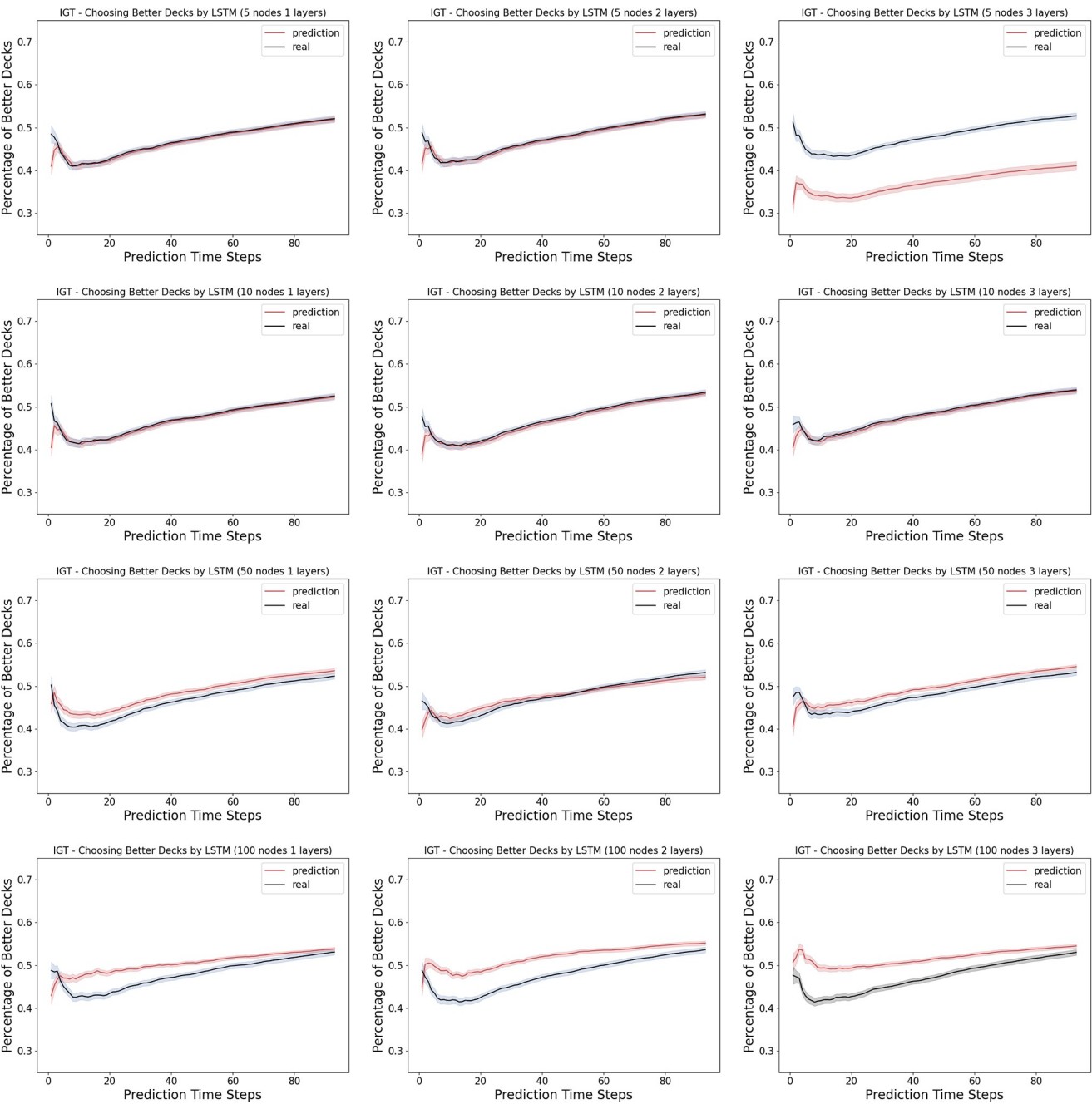

**Fig 5. Model complexity analysis in the Iowa Gambling Task (learning): Shown here are the percentage of choosing the better decks (i.e. the learning curves) computed from the prediction by the LSTM networks versus the real human data.** The first time step is given as the prior, and we record the prediction of the next 94 time steps in a 95-step Iowa Gambling Task game. The columns indicates the number of layers in the LSTM networks, ranging from 1, 2 and 3. The rows indicates the number of neurons in each layer of the LSTM networks, ranging from 5, 10, 50 and 100.

## 4 Discussion

As far as we are aware, this is the first work to predict the behavioral trajectories of the Iowa Gambling Task. In the Iterated Prisoner's Dilemma prediction task, [7] is the state-of-the-art with their logistic regression model. In our evaluations, our proposed LSTM model and

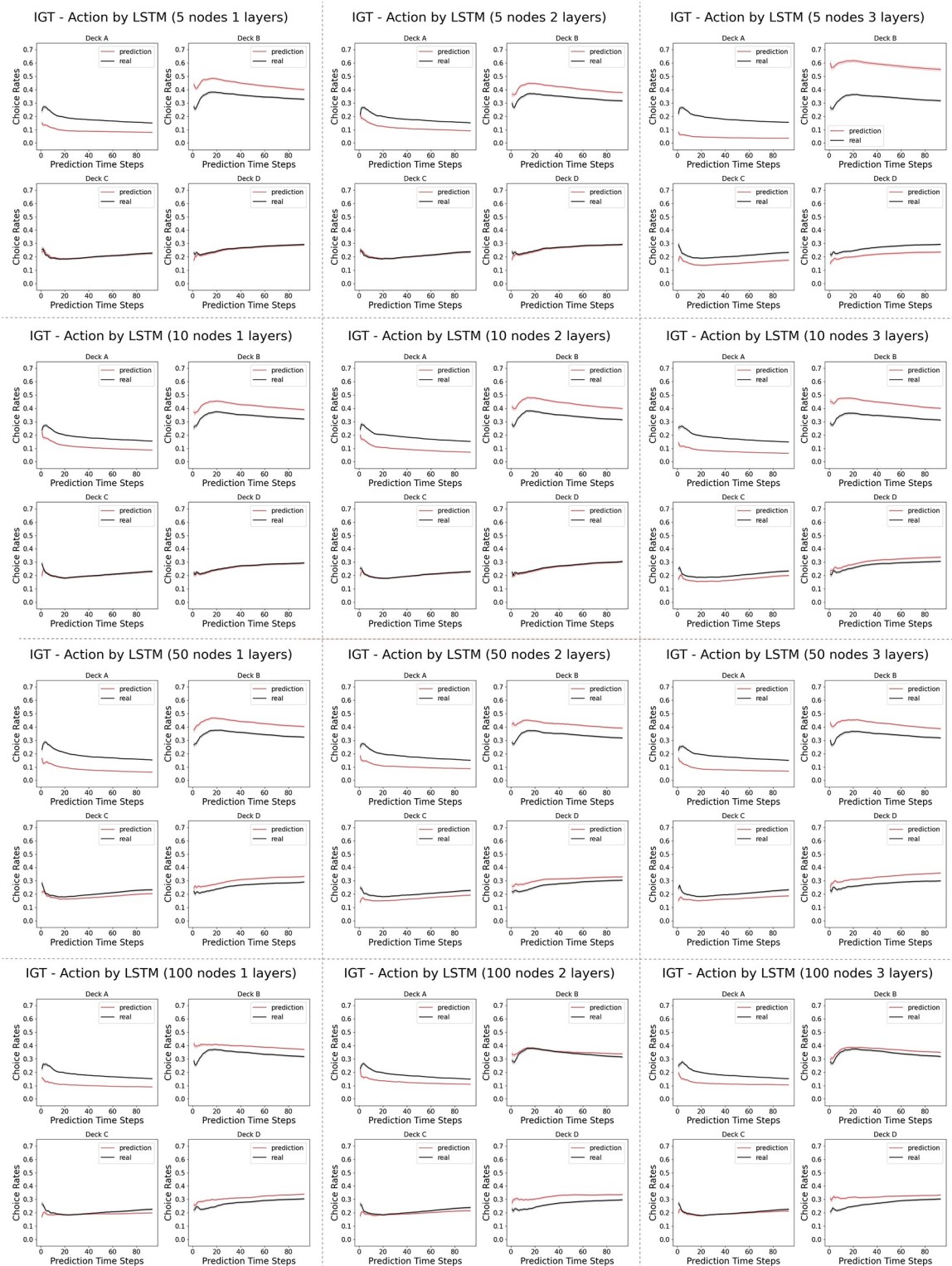

**Fig 6. Model complexity analysis in the Iowa Gambling Task (actions): Shown here are the percentage of choosing individual action arms computed from the prediction by the LSTM networks versus the real human data.** The first time step is given as the prior, and we record the prediction of the next 94 time steps in a 95-step Iowa Gambling Task game. The columns indicates the number of layers in the LSTM networks, ranging from 1, 2 and 3. The rows indicates the number of neurons in each layer of the LSTM networks, ranging from 5, 10, 50 and 100.

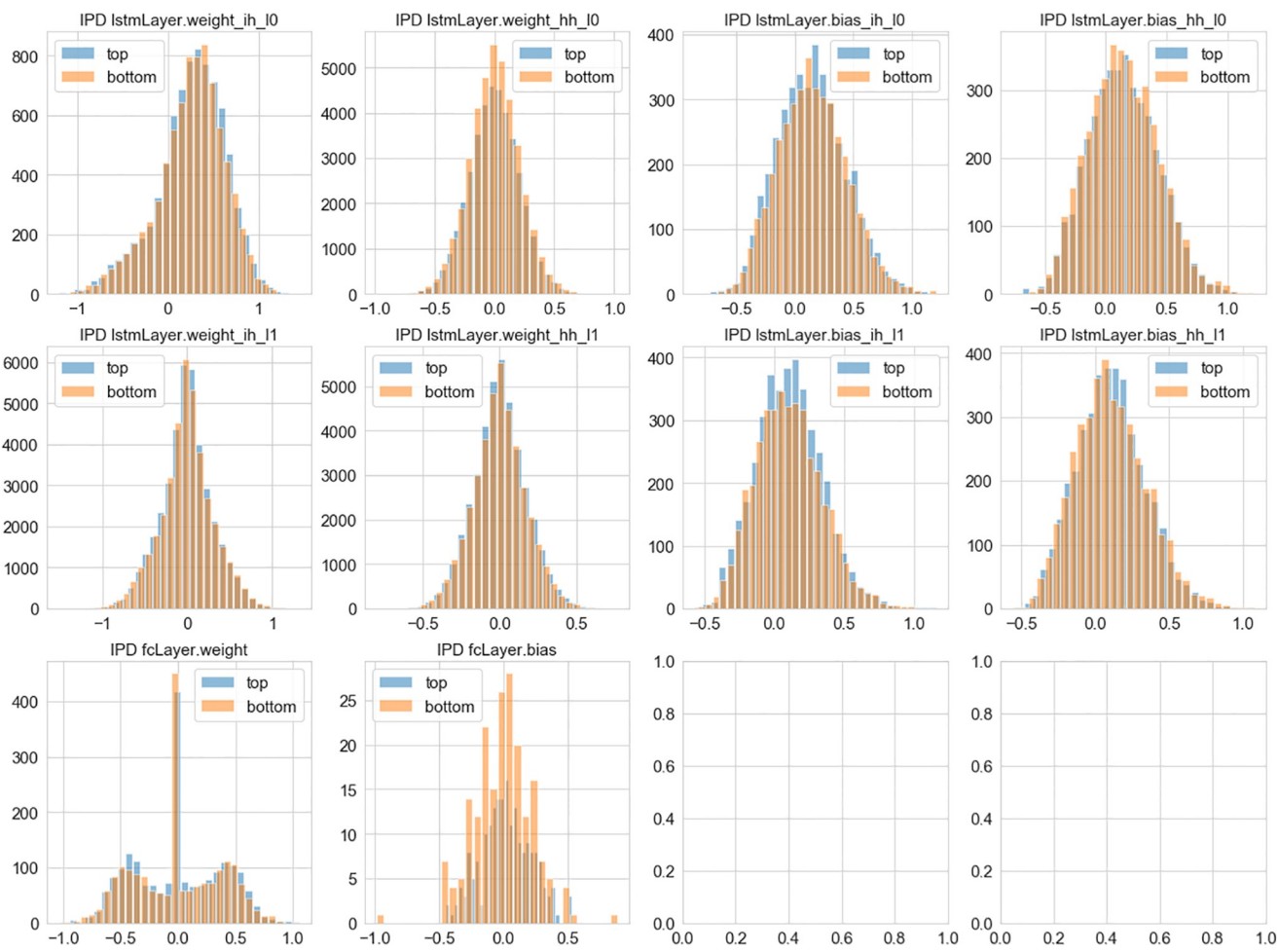

**Fig 7. Learned representations and game performance in the Iterated Prisoner's Dilemma:** Shown here are the distribution of weights learned from top 25% and bottom 25% performers selected based on the cumulative rewards in the historical records of human subjects playing the Iterated Prisoner's Dilemma.

autoregresssion baseline both significantly outperforms [7]. We discuss below the insights that our modeling approach provides.

## 4.1 Iterated Prisoner's Dilemma

The tendency to cooperation is an important subject of interest in Iterated Prisoner's Dilemma, because it characterizes the core trade-off between self-interest and social benefit. The cooperation rate is also the metric used by the state-of-the-art paper in predicting Iterated Prisoner's Dilemma sequences [7]. The prediction error by the three models offers several surprising observations: (a) Although the logistic regression model (the state-of-the-art) predicts the population-wide cooperation rate, it performs poorly at predicting individual actions; (b) Despite the significant overestimation of population cooperation rate, the autoregression model maintained a relatively low prediction error with a similar trend as the best model, our LSTM model; and (c) During the intermediate phase of iterations, the 4th or the 5th round, the prediction error appears to be the largest for the logistic regression model, but the smallest for the autoregression and LSTM models.

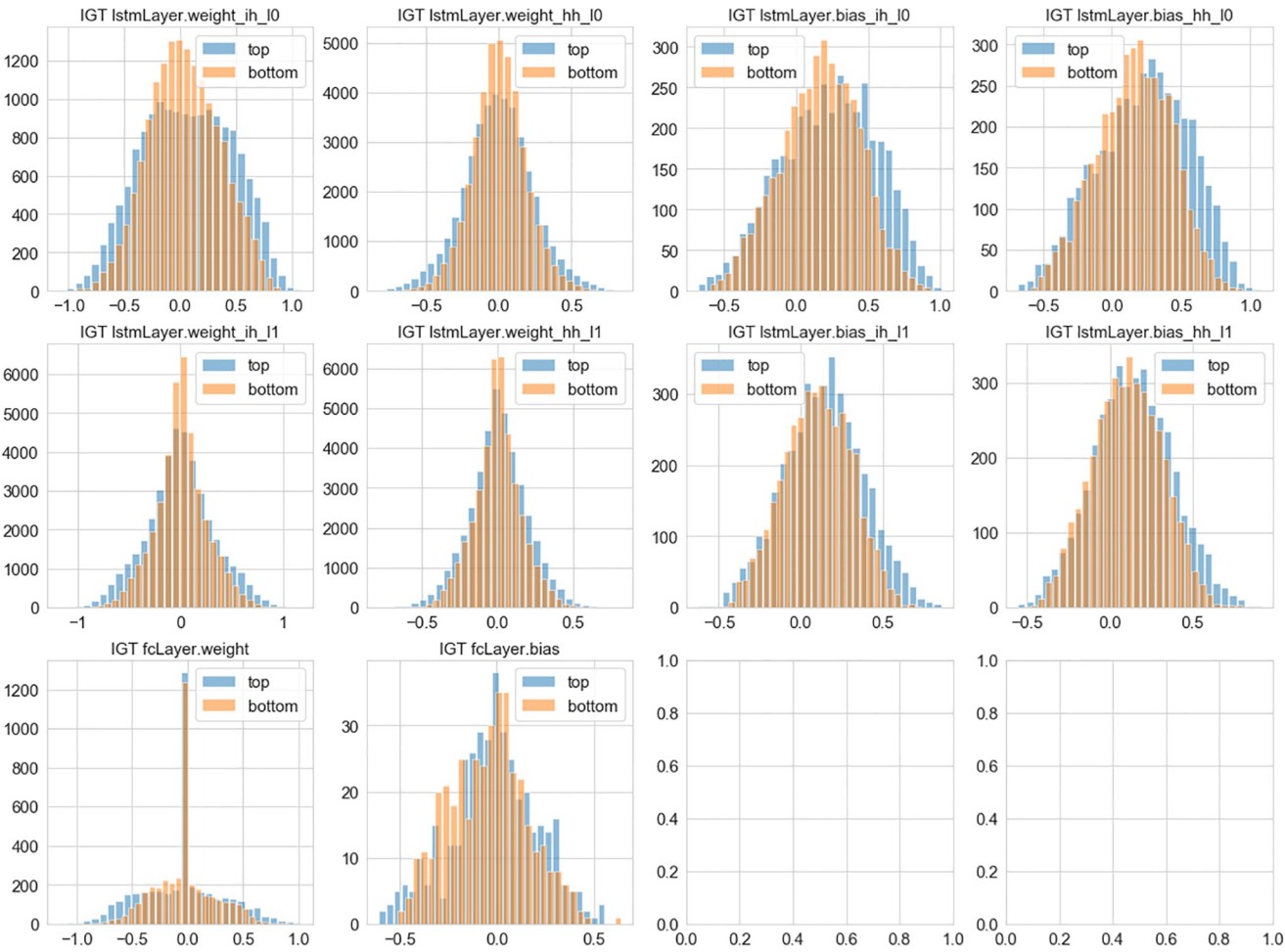

**Fig 8. Learned representations and game performance in the Iowa Gambling Task: Shown here are the distribution of weights learned from top 25% and bottom 25% performers selected based on the cumulative rewards in the historical records of human subjects playing the Iowa Gambling Task.**

### 4.2 Iowa Gambling Task

As shown in the payoff schemes from Table 2, there are two actions that are more preferable (giving more rewards) than the other two. In the psychology and neuroscience literature the percentage of choosing these two "better" actions is usually reported as a function of time, and used to characterize the learning progress of the human subjects. The deck A has a reward distribution of a five-modal distribution which is very wide, while the deck B has a rare probability of assigning a single large value which is very narrow. We suspect that the different distributions of the reward functions might be a reason behind the wrong prediction in our neural networks. This is further supported by the Fig 9, where the mean squared error (MSE) of the predictions of individual actions are presented for all 12 model architectures. We observe that smaller networks (narrower and shallower ones) predicts the action of choosing decks C and D perfectly (the good decks), while having some wrong predictions in decks A and B (the bad decks) better. We note that the reward distributions of the good decks are more Gaussian than the bad decks, whose reward distributions are more extreme, with very large values at very rare probabilities. In our case, the smaller networks can capture the Gaussian priors of the good decks very well, but don't have the expressive power to learn the more stochastic bad decks well as bigger networks do.

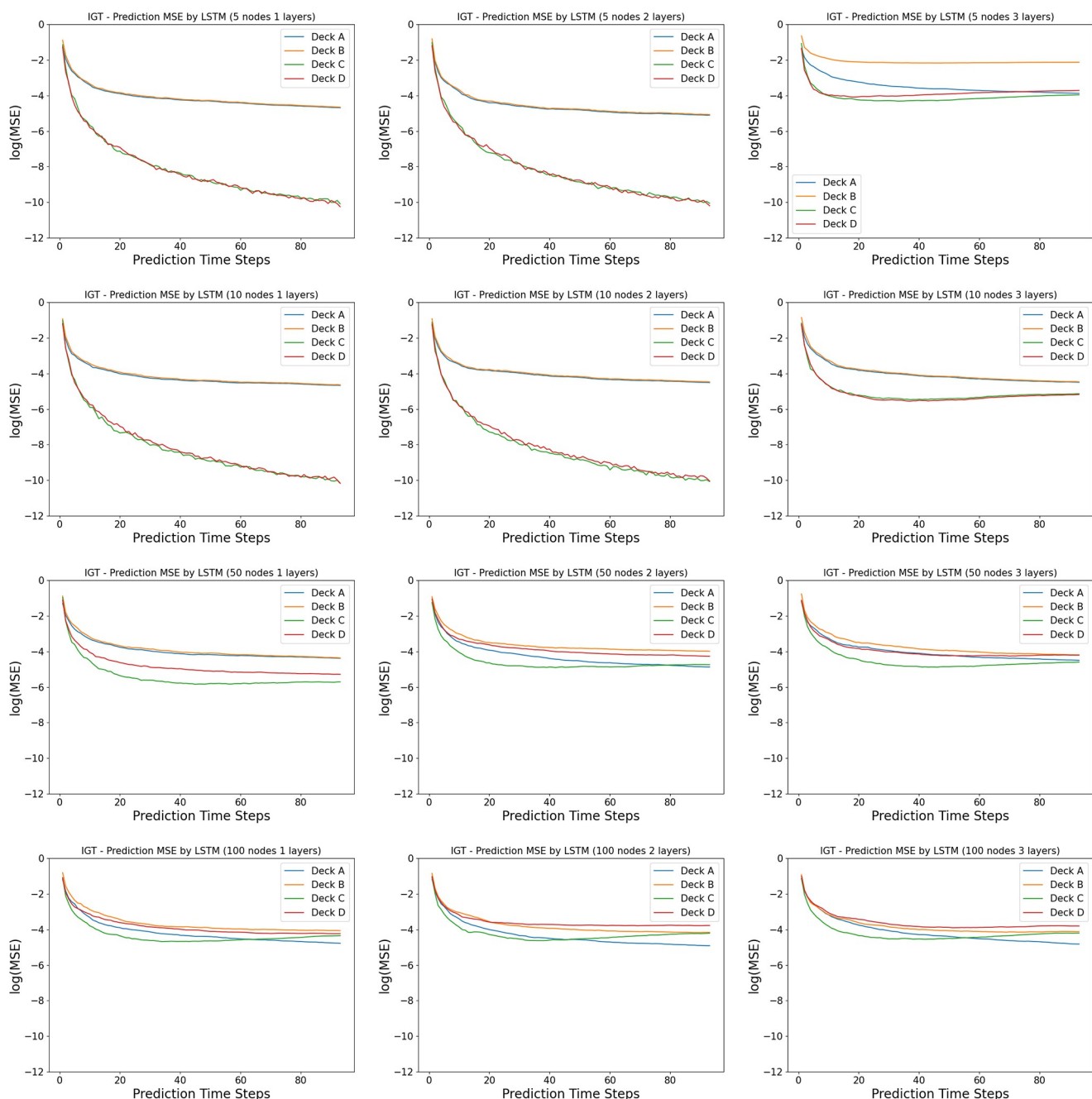

**Fig 9. Model complexity analysis in the Iowa Gambling Task (prediction error): Shown here are the mean squared error of indivual action arms between the prediction by the LSTM networks and the real human data.** The first time step is given as the prior, and we record the prediction of the next 94 time steps in a 95-step Iowa Gambling Task game. The columns indicates the number of layers in the LSTM networks, ranging from 1, 2 and 3. The rows indicates the number of neurons in each layer of the LSTM networks, ranging from 5, 10, 50 and 100.

## 4.3 Prediction of individual trajectories

In both cases, we observe that a model that predicts the cooperation rate well doesn't guarantee it captures the correct action strategies used by each individual trajectories. Despite a comparable performance in predicting the overall trend of the learning progress, the capability of a

prediction model to capture the action strategy during each individual game is a more challenging objective. For instance, it is possible for a bad prediction model to forecast every individual trajectories incorrectly given their corresponding heterogeneous prior history, while maintaining a perfect prediction for the composition of actions in the population sense. By examining the prediction errors and their corresponding individual action dimension at each time step, we can parse out more details about the strategies taken by different agents. For instance, a closer investigate to the individual action dimension reveals a difference in prediction strategies by the LSTM and the autoregression model in the Iterated Prisoner's Dilemma, but not in the Iowa Gambling Task. In the case of the Iterated Prisoner's Dilemma, the LSTM network not only predicts the overall cooperation rate better, but also effectively learns to mimic the overall learning trend of the Iowa Gambling Task in each decks. If we compare the Iowa Gambling Task with the Iterated Prisoner's Dilemma, we observe that the information asymmetry caused by the unknown human opponent complicates the prediction task in the Iterated Prisoner's Dilemma, such that the prediction error doesn't follow a monotonic decrease when the observation window increases, as we observe in the Iowa Gambling Task. A possible explanation is that predicting the Iterated Prisoner's Dilemma is a much more difficult task than predicting the Iowa Gambling Task due to its additional complications from the multi-agent and social dilemma settings. The clear advantage of the LSTM model over the baseline in this task, demonstrated the merit that the recurrent neural network better captures realistic human decision making and offers reliable prediction of individualized human behaviors.

## 4.4 Model complexity analysis

The results suggest that, despite being a multi-agent game, Iterated Prisoner's Dilemma has a much simpler strategy to learn from. Unlike the Iterated Prisoner's Dilemma, we observe that in the Iowa Gambling Task, the LSTM networks ranging from 1 layer of 5 neurons to 3 layers of 100 layers differ in their similarity of the learning curve in choosing the better actions to those of the corresponding human data. More specifically, we observe that the human data usually have a dip in the early rounds before catcH2ng up, wH2le the wider and deeper networks tend to adopt a simpler learning curve with a smaller dip.

## 4.5 Weight distribution in good and bad performers

We notice that the wider distribution of weights, significant for the Iowa Gambling Task and marginal but pointing in the same direction for the Iterated Prisoner's Dilemma, is suggestive of possible interpretations. One interesting possibility is that the better performers represent a larger number of alternative solutions wH2ch may be encompassed witH2n the expressivity of the LSTM; tH2s hypothesis is of course speculative for the moment, but we believe it may be eventually tested with further experimentation. Moreover, our analysis of LSTM's biases and weights points to possible ways for describing alternative solution strategies leading to significantly different outcomes.

## 4.6 Potential implications

Good predictors of human decision-making trajectories can help government develop better resource allocation programs, help companies develop better recommendation systems, and help clinicians develop intervention plans for mental health treatments. As a comparison, reinforcement learning models are good at mechanistically capturing the psychological activities. Our prior work [17] provides a negative result for using reinforcement learning models to predict human decision making, wH2ch suggests the necessity of an additional behavioral

predictor model is in demand, and can serve useful purposes in many real-world application that involves predictive modeling. We observe in tH2s predictive modeling investigation a possibility that these behavioral predictors might capture characteristics of human decision making process that are missed in reward-driven models, partially because in behavioral experimental settings, the reward representations, usually monetary, can be an over-simplification of the complex underlying mechanisms of human minds.

As the first attempt to utilize the recurrent neural networks to directly predict human action sequences in these behavioral tasks, our approach matches existing baselines in predicting both the population trends and the individual strategies, in the Iowa Gambling task, and then significantly outperforms the state-of-the-art in the Iterated Prisoner's Dilemma task. We find the latter particularly noteworthy given that Iterated Prisoner's Dilemma is a cognitively more complex task, as it involves multiple agents trying to predict each other's behavior. Next steps include extending our evaluations to human behavioral trajectories in other sequential decision making environments with more complicated and mixed incentive structure, such as Diplomacy, Poker and chess playing, as well as efforts to implement alternative recurrent models more readily amenable to interpretation from the neuroscientific and psychological perspectives.

## Author Contributions

**Conceptualization:** Baihan Lin, Djallel Bouneffouf, Guillermo Cecchi.

**Data curation:** Baihan Lin.

**Formal analysis:** Baihan Lin.

**Investigation:** Baihan Lin.

**Methodology:** Baihan Lin, Djallel Bouneffouf, Guillermo Cecchi.

**Project administration:** Baihan Lin, Djallel Bouneffouf, Guillermo Cecchi.

**Resources:** Baihan Lin, Djallel Bouneffouf, Guillermo Cecchi.

**Software:** Baihan Lin.

**Supervision:** Baihan Lin, Guillermo Cecchi.

**Validation:** Baihan Lin.

**Visualization:** Baihan Lin.

**Writing – original draft:** Baihan Lin.

**Writing – review & editing:** Baihan Lin, Djallel Bouneffouf, Guillermo Cecchi.

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
