## [Decision Letter · Decision Letter 0]

4 Mar 2022

PONE-D-21-36573Predicting human decision making in psychological tasks with recurrent neural networksPLOS ONE

Dear Dr. Lin,

Thank you for submitting your manuscript to PLOS ONE. After careful consideration, we feel that it has merit but does not fully meet PLOS ONE’s publication criteria as it currently stands. Therefore, we invite you to submit a revised version of the manuscript that addresses the points raised during the review process.

We look forward to receiving your revised manuscript.

Kind regards,

Luo-Luo Jiang, Ph.D.

Academic Editor

PLOS ONE

Journal Requirements:

This work was financially 333

supported in part by the Systems Biology Fellowship awarded by Columbia University 334

and the research training grants awarded by the National Science Foundation and the 335

National Institutes of Health. 

Reviewers' comments:

Reviewer's Responses to Questions

**Comments to the Author**

1. Is the manuscript technically sound, and do the data support the conclusions?

Reviewer #1: Partly

2. Has the statistical analysis been performed appropriately and rigorously? 

Reviewer #1: Yes

3. Have the authors made all data underlying the findings in their manuscript fully available?

Reviewer #1: Yes

4. Is the manuscript presented in an intelligible fashion and written in standard English?

Reviewer #1: Yes

5. Review Comments to the Author

Reviewer #1: I cannot say this manuscript was well-written.

1. Using exactly the same sentences presented in the text to describe the table won't be a wise choice.

2. Discussions should not be presented in Results section.

3. Some of the vague sentences hinted a stiff translation from other language.

4. Generally, the work done by the authors was not enough to address the significance of the idea.

I will recommend the authors to deeply revise the manuscript before sending it back for consideration of publication.

6. PLOS authors have the option to publish the peer review history of their article (what does this mean?). If published, this will include your full peer review and any attached files.

Reviewer #1: No

---

## [Author Response · Author response to Decision Letter 0]

14 Apr 2022

We thank the reviewer for the careful read and constructive feedback. As you can see from our difference-highlighted revision manuscript, we have fully taken into account your helpful suggestions and significantly revised our manuscript. The changes are highlighted in detail in the attached file (difference_HumanLSTM.pdf) and summarized as follows: 

- We modified the abstract to reflect a more explicit, constrained and solid description of what we have accomplished.

- To better address the significance of our idea, we modified the last paragraph of the introduction to address the main objectives.

- The structure of the paper is now reorganized in two ways: we merged the "Background" into the "Materials and Methods", and separate the "Discussion" out of the "Results", per your suggestion.

- We changed the description of the tables to distinguish it from the main text.

- We introduced more details about the data collection for both the Iowa Gambling Task and Iterated Prisoner's Dilemma into the "Methods and Materials" section.

- The description of the recurrent neural network was moved forward, because it is the main methodological component.

- We significantly rewrote our "Discussion" section such that core conclusions are well structured and connected to the conclusions. Main contributions are reiterated in the front and the back. The interpretations related to individual experiments are discussed in more details. 

We hope these improvements in the writing are able to better address the significance of our idea. We thank you again for your time and efforts for reviewing our manuscript.

---

## [Editor Report · Decision Letter 1]

19 Apr 2022

Predicting human decision making in psychological tasks with recurrent neural networks

PONE-D-21-36573R1

Dear Dr. Lin,

We’re pleased to inform you that your manuscript has been judged scientifically suitable for publication and will be formally accepted for publication once it meets all outstanding technical requirements.

Kind regards,

Luo-Luo Jiang, Ph.D.

Academic Editor

PLOS ONE
---

## [Editor Report · Acceptance letter]

11 May 2022

PONE-D-21-36573R1 

Predicting human decision making in psychological tasks with recurrent neural networks 

Dear Dr. Lin:

I'm pleased to inform you that your manuscript has been deemed suitable for publication in PLOS ONE. Congratulations! Your manuscript is now with our production department. 

Kind regards, 

on behalf of

Dr. Luo-Luo Jiang 

Academic Editor

PLOS ONE